# WS-AM: Weakly Supervised Attention Map for Scene Recognition

**Shifeng Xia [1], Jiexian Zeng [1,2], Lu Leng [1,\*] and Xiang Fu [1]**

[1]   School of Software, Nanchang Hangkong University, Nanchang 330063, China; 1716083500006@stu.nchu.edu.cn (S.X.); zengjx58@163.com (J.Z.); fxfb163@163.com (X.F.)

[2]   Science and Technology College, Nanchang Hangkong University, Gongqingcheng 332020, China

\*   Correspondence: leng@nchu.edu.cn

**Abstract:** Recently, convolutional neural networks (CNNs) have achieved great success in scene recognition. Compared with traditional hand-crafted features, CNN can be used to extract more robust and generalized features for scene recognition. However, the existing scene recognition methods based on CNN do not sufficiently take into account the relationship between image regions and categories when choosing local regions, which results in many redundant local regions and degrades recognition accuracy. In this paper, we propose an effective method for exploring discriminative regions of the scene image. Our method utilizes the gradient-weighted class activation mapping (Grad-CAM) technique and weakly supervised information to generate the attention map (AM) of scene images, dubbed WS-AM—weakly supervised attention map. The regions, where the local mean and the local center value are both large in the AM, correspond to the discriminative regions helpful for scene recognition. We sampled discriminative regions on multiple scales and extracted the features of large-scale and small-scale regions with two different pre-trained CNNs, respectively. The features from two different scales were aggregated by the improved vector of locally aggregated descriptor (VLAD) coding and max pooling, respectively. Finally, the pre-trained CNN was used to extract the global feature of the image in the fully- connected (fc) layer, and the local features were combined with the global feature to obtain the image representation. We validated the effectiveness of our method on three benchmark datasets: MIT Indoor 67, Scene 15, and UIUC Sports, and obtained 85.67%, 94.80%, and 95.12% accuracy, respectively. Compared with some state-of-the-art methods, the WS-AM method requires fewer local regions, so it has a better real-time performance.

**Keywords:** convolution neural network; scene recognition; vector of locally aggregated descriptor; weakly supervised attention map

## 1. Introduction

Scene recognition, as a sub-problem of image recognition, has attracted increasing attention. It has important applications in robotics, intelligent security, driving assistant technique, and human-computer interaction, etc. However, scene recognition is quite different from general object recognition:

- Scene images, especially indoor scene images, commonly contain a large number of objects and a complex background;
- Human ability in scene recognition is much lower than that in object recognition;
- The number of datasets of scene recognition is much less than that of object recognition.

There are also several difficulties in scene recognition, such as variances of illumination, scale, and so on. The variability and difference of scene content lead to inter-class similarity and intra-class variation. Figure 1 shows some difficulties in scene recognition.

The focus of scene recognition is to extract more robust and generalized features, including hand-crafted features and learning-based features. Traditional scene recognition methods generally use hand-crafted features, e.g., oriented texture curves (OTC) [1], census transform histogram (CENTRIST) [2], histogram of oriented gradient (HOG) [3], and scale-invariant feature transform (SIFT) [4]. Hand-crafted features are constructed based on image color, texture, structure, and other information. They have no semantic information and are difficult to use in complex scene recognition. With the wide use of deep learning in computer vision, learning-based features have been applied to scene recognition. Convolutional neural network (CNN) is a typical representative of learning-based features [5–8]. Latent feature representation containing high-level semantic information can be learnt from large-scale data without human intervention. Even though the CNN features perform well in scene recognition [9], they still use global information while ignoring local information, and cannot satisfactorily solve between-class similarity and within-class difference.

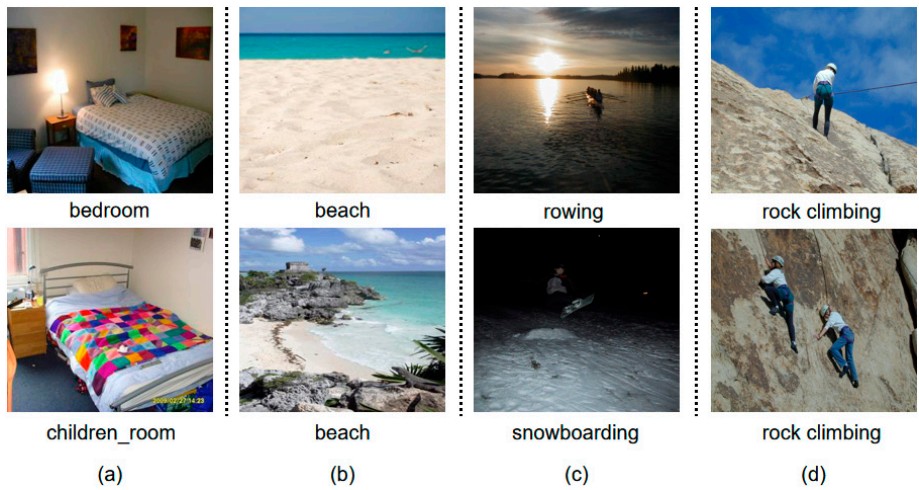

**Figure 1.** Some difficulties in scene recognition. (**a**) inter-class similarity. (**b**) intra-class variations. (**c**) illumination problem. (**d**) shooting angle problem.

Intuitively, the images of scene recognition are different from the general images of object recognition. Many scene images contain a large number of objects, especially indoor scenes, and have a complex background, which brings severe difficulties for feature extraction. Many CNN-based methods extract features from local regions at different scales and complement global representation; however, they do not sufficiently consider the relationship between the local region and the context of the scene. Many extracted local regions are redundant and degrade the classification results of scenes. The scene images of different categories often contain the same and similar object regions, while the scene images of the same category probably contain very different object regions. In this paper, we focus on the discriminative regions in scene images. The feature extraction of discriminative regions can effectively solve the problems of between-class similarity and within-class difference. Figure 2 shows the images of two categories ('bedroom' and 'children's room') of MIT indoor 67 dataset [10]. It can be seen from each column that the samples of different scene categories are very similar. Significant intra-class differences are remarkable in each row, which is caused by different backgrounds, objects, and angles. In order to achieve significant recognition results, a suitable way is to find discriminative region blocks that are good representations helpful to classification.

In this paper, we propose a weakly supervised attention map (WS-AM) method, which uses the gradient-weighted class activation mapping (Grad-CAM) [11] technique to obtain a small-scale attention map (AM) for each image. WS-AM uses the maximum output value information of the last fully-connected (fc) layer of CNN, but the image-level label information is absent, so it can be considered as weakly supervised. The regions with large local mean and large local center value in AM correspond to the regions of the original image that have strong discriminative power, while the others

correspond to the redundant regions in the original image. The features are extracted from multi-scale discriminative regions per image. The features in small-scale regions are extracted in the softmax layer using CNN that is pre-trained on the ImageNet dataset [12] (i.e., ImageNet-CNN), and then they are coded by improved vector of locally aggregated descriptor (VLAD) [13] and normalized with L2-normalization. The features of large-scale regions are extracted in the softmax layer using CNN that is pre-trained on the Places365 dataset [9] (i.e., Places365-CNN), and then they are aggregated by max pooling. In order to obtain the global feature of the image, we use Places205-CNN (i.e., CNN pre-trained on Places205 dataset) [9] to extract the feature vector in the first fc layer (i.e., fc6 layer), and they are normalized with L2-normalization. Finally, the three feature vectors are concatenated to form the final image representation. In order to verify the effectiveness of WS-AM, the experiments were carried out on three datasets and achieved good performance.

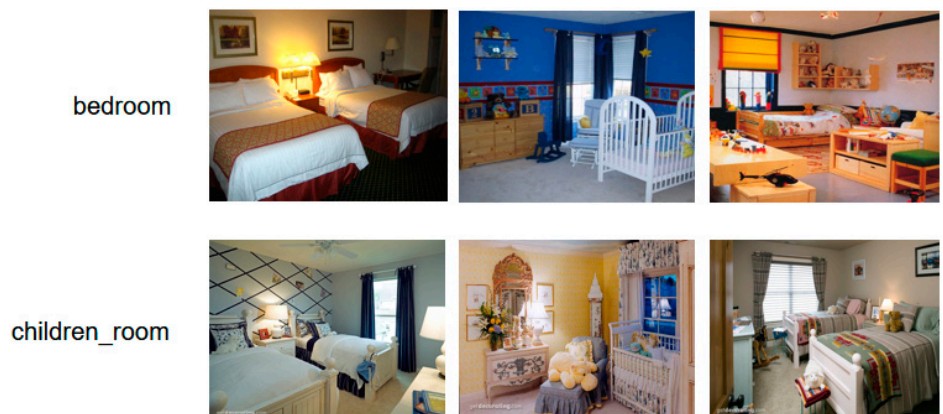

**Figure 2.** Images of two categories from the MIT indoor 67 dataset. Each row shows the difference within the class. Each column shows the similarity between classes.

The remainder of this paper is organized as follows. The related works are reviewed in Section 2. Section 3 introduces our method, including the pipeline and details of the whole algorithm. Sections 4 and 5 introduce the experiments and analysis in detail. Finally, we summarize the whole work in Section 6.

## 2. Related Work

In this section, the related work is briefly reviewed, including scene representation, discriminative region discovery, feature coding, and scene classification.

### 2.1. Scene Representation

In traditional scene recognition, hand-crafted features are widely used because they are relatively simple and have low computational cost. Traditional scene recognition can be divided into the following steps: extract patches, represent patches, encode features, and pool features. In the patch representation, the features, such as SIFT, HOG, and speeded-up robust features (SURF) [14], are extracted from local regions. Effective hand-crafted features can not only depict the texture characteristics but also reflect the deep structure information. The Bag of Features (BOF) model based on SIFT feature has been widely used in scene recognition, but the lack of location information makes it difficult to use in a complex scene. Lazebnik et al. [15] improved the BOF model based on SIFT feature and proposed the spatial pyramid matching (SPM) model, which achieved good results in scene recognition. HOG feature was initially used in pedestrian detection. Later, Felzenswalb et al. [16] proposed the deformable parts model (DPM) on the basis of HOG feature. Pandey et al. [17] improved the DPM and applied it to large-scale scene image recognition. After clustering and coding of local features of scene images, pooling operations are needed. Max pooling and average pooling are commonly used in pooling

operations. The experimental results of Yang et al. [18] on several benchmark databases show that the effect of max pooling is better than that of average pooling.

Recently, CNNs have made prominent progress on computer vision, especially in image recognition. AlexNet [19] won the championship in the ImageNet image recognition competition in 2012. Since then, CNNs have made breakthroughs in object detection, semantic segmentation, and image generation. Benefiting from large-scale well-labeled datasets, more CNN structures have been proposed, such as VGGNet [20], GoogLeNet [21], and ResNet [22]. CNNs are also widely used in scene recognition. Zhou et al. [9] used CNNs to train and test on a new large-scale scene dataset Places and achieved great results. Although the global features extracted by CNN have achieved remarkable results in scene recognition, they only represent the global information, and ignore the local information. Shi et al. [5] recently proposed a novel approach which utilized the visually sensitive features combining with CNN features for scene recognition. Wang et al. [23] proposed a multi- resolution CNN structure to capture visual content and structure at multiple levels of images. Javed et al. [24] proposed a deep network structure, which uses the position relations of a group of objects to infer the scene category, and then establishes the semantic context model of the scene. Many methods do not train CNNs from scratch, but directly use the CNNs, namely Places205-CNN, Places365-CNN, and ImageNet-CNN, pre-trained on the three large datasets (i.e., Places205, Places365, and ImageNet) to extract features.

## 2.2. Discriminative Region Discovery

Local region information is very important for scene recognition, but current methods do not sufficiently focus on the discriminative region of the scene image. Some methos densely sampled local regions in a multi-scale way for scene images [6,25,26]. Dense sampling extracts all regions of the image, but it inevitably produces many redundant regions, most of which are in the background without objects or contain similar regions in different scene categories. Dense sampling also leads to high computational cost. Uijlings et al. [27] proposed a selective search method for generating a set of regions that are likely to contain objects. Selective search is a region proposal method and widely used in object detection. Intuitively, most scenes consist of many objects, so the region proposal method can be used to generate local regions containing objects. Wu et al. [8] used multi-scale combinatorial grouping (MCG) [28] to generate high-quality local regions for scene images. Javed et al. [24] utilized edge boxes [29] to extract image candidate regions, and feature maps of the same size can be generated by region of interest (RoI) pooling for the candidate regions. However, those unsupervised region proposal approaches fail to consider the relationship between object regions and scene categories, and still produce some redundant regions. Discriminative power analysis [30,31] can help judge whether the regions are discriminative or redundant.

Zhou et al. [32] proposed a method to generate class activation mapping (CAM) using the global average pooling (GAP) in CNNs. The CAM of a specific category represents the discriminative image region for identifying this category. CAM forces the CNN structure to include GAP, but some CNN structures do not have GAP, such as AlexNet and VGGNet. In order to solve this problem, Selvaraju et al. [11] put forward the Grad-CAM technique, which uses the gradient of the interested class to propagate back to the convolutional layer to generate a coarse localization map. It highlights the discriminative regions to predict the interested category. Recently, the attention mechanism has been widely used in computer vision tasks, such as fine-grained image recognition [33–35], scene text recognition [36–38], and so on. Fu et al. [33] proposed a novel recurrent attention convolutional neural network (RA-CNN) for fine-grained image recognition. RA-CNN learns discriminative region attention and region-based feature representation in a recursive way, without the use of any bounding box annotation information. Gao et al. [37] introduced a text attention module in the text feature extraction process to focus on text regions and avoid background noise. These works utilize attention modules to capture category-specific objects and parts. Lorenzo et al. [39] proposed a new attention-based CNN for selecting bands from hyperspectral images. This method uses gating mechanisms to obtain the most informative regions of the spectrum. Attention mechanisms are also widely used in other network

structures, e.g., long short-term memory (LSTM) [40] and gated recurrent (GRU) [41] neural networks. Vaswani et al. [42] proposed a new simple network architecture based on attention mechanisms, called the Transformer. The Transformer has achieved outstanding results on two machine translation tasks. Inspired by these works, we apply the attention module to scene recognition. Our Grad-CAM based method has obvious advantages:

- Our method uses pre-trained CNN as the backbone network of the attention module, instead of training from scratch or fine-tuning;
- Different from other attention modules that select a fixed number of regions per image, our method obtains an adaptive number of regions for each image, which is more conducive to scene recognition;
- Different from other attention modules, our AM does not use image-level label information.
- Compared with other methods, our method is simpler and does not require adding new components to the network structure to drive the attention mechanism.

### 2.3. Feature Coding

In traditional scene recognition, clustering and coding local features are needed to obtain image embedding. The feature coding methods can be mainly divided into two types: global coding and local coding. Global coding is usually used to estimate the probability density distribution of features, while local coding is used to describe each feature. Typical feature coding includes bag of visual words (BoVW) [43,44], fisher vector (FV) [45,46], VLAD, and salient coding (SC) [47]. FV coding uses the Gaussian mixture model (GMM) to estimate the distribution of features. GMM consists of weights, means, and covariance matrices of several Gaussian distributions, each of which reflects a feature pattern. As a simplification of FV, VLAD calculates the residuals between the features and the nearest neighbor visual dictionary. VLAD takes into account the value of each dimension of features and describes the local information of images in a more detailed, simple, and effective way, so it has been widely used in scene recognition.

Feature coding is also important for scene recognition based on deep learning. Many traditional feature coding methods have been improved to be more suitable for deep learning. Dixit et al. [6] proposed semantic FV for scene recognition by combining the local features extracted from traditional FV and CNNs. Khan et al. [48] proposed Deep Un-structured Convolutional Activation (DUCA), which extracts the features from middle-level regions of images through CNNs and encodes them according to their association with the codebook of representative regions of scenes.

### 2.4. Scene Classification

There are mainly two types of classifiers for scene classification: discriminative models and generative models. The learning of the discriminative model is a conditional probability, which mainly focuses on the classification boundary of data. The discriminative model seeks the optimal separating hyperplane between different categories and reflects the difference between the different types of data. The advantages of the discriminative model are as follows:

- It can distinguish well the differences between categories;
- It is suitable for the identification of more categories;
- It is relatively simple and easy to understand.

However, the discriminative model does not reflect well the characteristics of the data. Commonly used discriminative models include k-nearest neighbor (KNN), logistic regression (LR), and support vector machine (SVM). In particular, SVM is widely used in scene recognition [6,8,25].

Different from the discriminative model, the generative model learns the joint probability distribution, which represents the distribution of data from a statistical perspective and can reflect the similarity of similar data. The generative model gives the joint probability density, which contains more

information, and its training speed is much faster than the discriminative model. However, the learning and calculation process of the generative model is complex, and the accuracy of the classification problem is lower than that of the discriminative model. The widely used generative models include the naive Bayesian model (NBM), hidden Markov model (HMM), and GMM.

## 3. Proposed Method

In order to distinguish one scene category from another, the most effective approach is to obtain category-specific objects or regions. Although many methods can obtain regions containing objects, many objects are not category-specific. Some regions contain objects that are common in different scenes, which introduce noise for feature extraction. To avoid common object regions, we propose a Grad-CAM based method to capture regions that only contain category-specific objects. The proposed method can be divided into two parts: WS-AM and scene representation. Figure 3 shows the main flow of our method. First, Grad-CAM is employed to generate AM for the input image, in which weakly supervised information (i.e., the maximum output value of the last fc layer) is used. The regions with large local mean and large local center value in AM correspond to the regions with strong discriminative power in the images. Second, we extract the multi-scale CNN features from these discriminative regions. Different scale regions are input to different pre-trained networks (i.e., ImageNet-CNN and Places365-CNN) and the feature vectors are extracted in the softmax layer. The features extracted from small-scale regions are aggregated by improved VLAD coding and normalized by L2-normalization. While max pooling is used for the features extracted from large-scale regions. The global feature is extracted in the first fc layer (i.e., fc6 layer) on Places205-CNN and normalized by L2-normalization. Finally, the three extracted features are concatenated to form the final image representation.

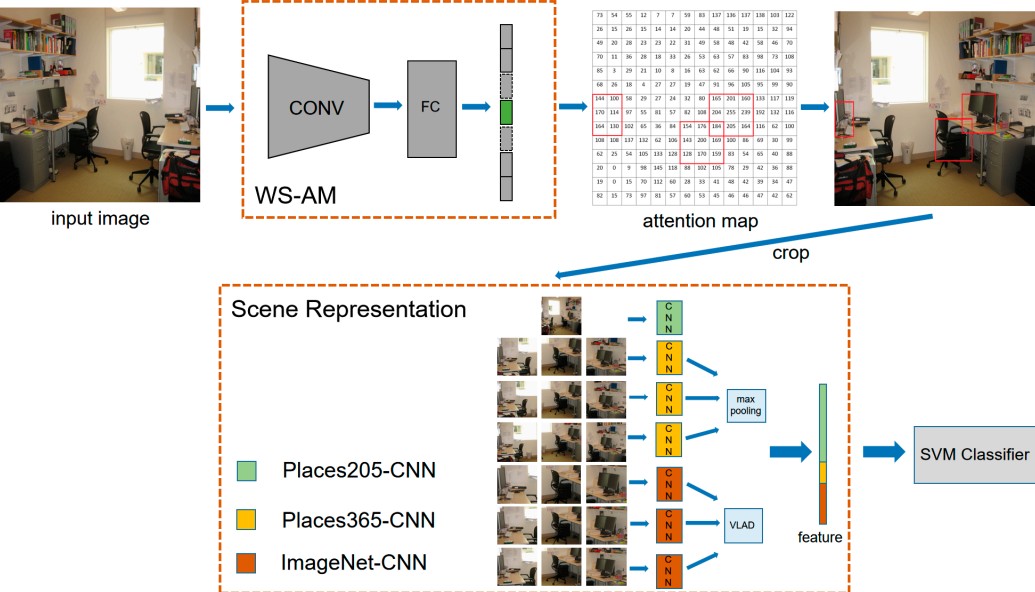

**Figure 3.** The framework of our method. The framework can be divided into two parts: WS-AM and scene representation.

### 3.1. Weakly Supervised Attention Map

WS-AM is used for discovering discriminative regions in scene images. Scene recognition is different from general object recognition, which is composed of complex background and various objects. Inspired by the work of Grad-CAM on the visual interpretation of CNNs, we use this method to generate the AM for each image. The backbone network for Grad-CAM is VGGNet pre-trained on the Places205 dataset, i.e., Places205-VGGNet. We do not use Place205-VGGNet to fine-tune the datasets, so the image-level label information is not used. Instead, the maximum output value in

the last fc layer of Places205-VGGNet is used as the backpropagation information to generate AM, which can be considered as weakly supervised. The gradient information is back-propagated to the last convolution (conv) layer to calculate the importance of each neuron to the final classification.

As shown in Figure 4, the input image $I$ is resized into the size of $224 \times 224$ and propagated forward through the CNN to obtain the output value of the last fc layer. The maximum output value $S$ is back-propagated to calculate the gradient of the feature maps $A$ at the last conv layer, i.e., $\partial S / \partial A$. $A^k$ represents the $k_{th}$ feature map of $A$, so the gradient of $A^k$ is $\partial S / \partial A^k$. Then the gradients of $k_{th}$ feature map are averaged to obtain the neuron importance weight $\alpha^k$ as follows:

$$\alpha^k = \frac{1}{Z} \sum_i \sum_j \frac{\partial S}{\partial A_{ij}^k} \qquad (1)$$

where $Z$ denotes the size of the $k_{th}$ feature map, which is $14 \times 14$. The weight $\alpha^k$ represents the local linearization of the feature map $A^k$, and also indicates the importance of the $k_{th}$ feature map to the maximum output value $S$. We take the sum of weighted feature maps, and by the activation function ReLU to obtain AM:

$$AM = \text{ReLU} \left( \sum_k \alpha^k A^k \right) \qquad (2)$$

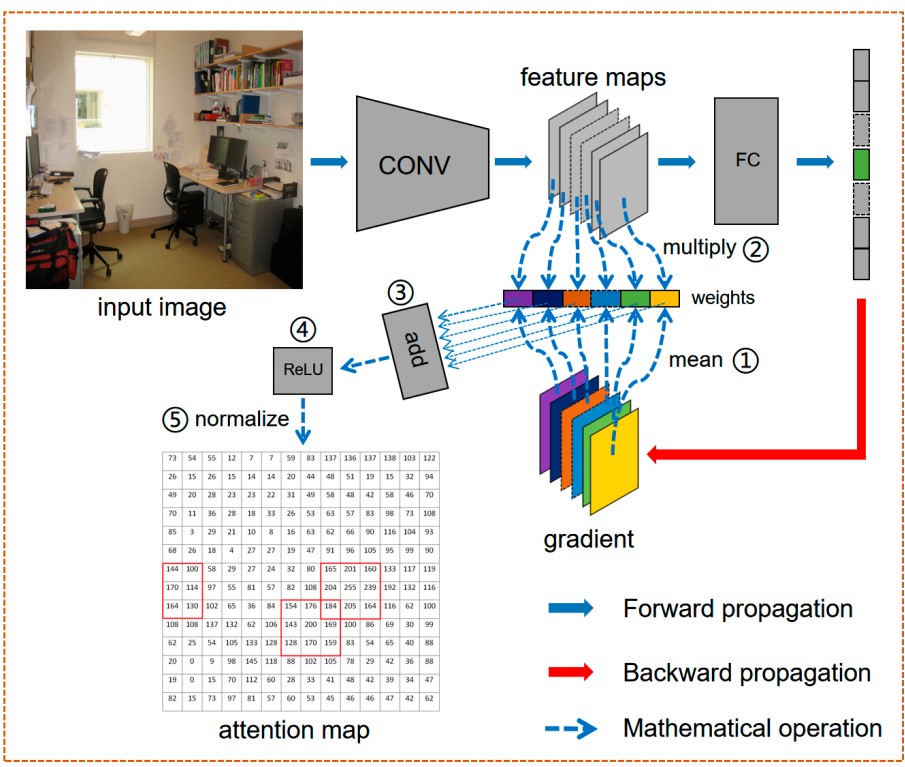

**Figure 4.** The pipeline of WS-AM. We use weakly supervised information to generate the attention map for each image.

ReLU is applied to linear combinations of feature maps and weights because we are only interested in those features which have a positive impact on the maximum output value, and the intensity of those feature pixels should be increased to enhance the category with the maximum output value [11]. The backbone network we used for Grad-CAM is VGGNet, so the AM size is $14 \times 14$. In order to facilitate calculation and visualization, the values in AM are normalized to the range of (0, 255). If the AM is up-sampled to the input image size (i.e., $224 \times 224$), each pixel value in the AM represents the importance of the corresponding pixel in the input image to the final classification result. Figure 5

shows four Grad-CAM visualization examples of the VGGNet pre- trained in the Places205 dataset. We only use weakly supervised information, but the discriminative region for each image is consistent with a human attention mechanism.

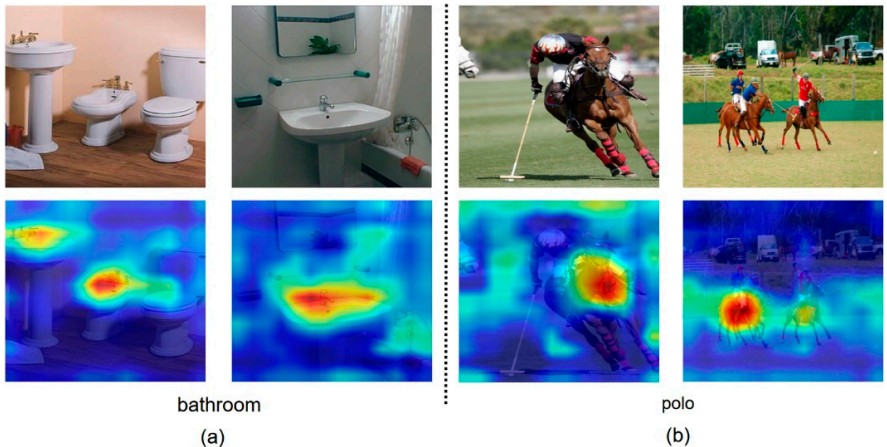

**Figure 5.** Some examples of Grad-CAM visualization.

A sliding window with $3 \times 3$ size and 1 stride is used to slide AM. In order to obtain the discriminative regions, two strategies are employed:

- The mean of the 9 numbers in the window is greater than the AM mean;
- The value of the window center is the maximum value of the 9 numbers and needs to be greater than the threshold value.

If both strategies are satisfied, the corresponding regions in the original image are considered as discriminative regions. The first strategies eliminate the exception window of AM in which the center value is larger than the threshold but other values are too small. Each discriminative region of the original image is cropped in the size of $s \times s$ in a multi-scale way, where $s \in \{64, 80, 96, 112, 128, 144\}$. Then, we resize each scale region into the size of $224 \times 224$ in order to adapt to the input size of VGGNet. Intuitively, small-scale regions ($s = 64, 80, 96$) contain 'object', while large-scale regions ($s = 112, 128, 144$) contain 'scene', so ImageNet-CNN and Places365-CNN are respectively used to extract local features.

*3.2. Improved Vlad*

In general, VLAD coding first carries out k-means cluster for local features and then calculates the accumulated residuals between the local features and their nearest neighbor cluster centers, and finally forms the image embedding as the local representation through pooling. VLAD has two shortcomings:

- It only considers the residual with the nearest neighbor cluster center;
- The encoded feature dimension is too high.

Furthermore, the number of small-scale regions is unbalanced, so the general VLAD coding cannot work well. To solve the above problems, VLAD coding is improved.

The feature vectors $l = [l_1, \ldots, l_j, \ldots, l_M]$ of the small-scale local regions in each image are non-Euclidean, so they are difficult to carry out for VLAD coding, $M$ denotes the number of the small-scale local features of each image. Natural parameterization is used to transform these feature vectors into linear Euclidean space as follows:

$$v_j = \sqrt{l_j} \tag{3}$$

where $v_j$ is the transformed feature vector. The conversion from non-Euclidean space to linear Euclidean space is more conducive to VLAD coding. Mini Batch k-means method clusters all

small-scale local features, and obtains codebook with $k$ cluster centers $c = [c_1, \ldots, c_i, \ldots, c_k]$. For the local features $v = [v_1, \ldots, v_j, \ldots, v_M]$ of each image, we calculate the residuals between each feature $v_j$ and all clustering centers. Then, the residuals of each cluster center are aggregated, and the formula is as follows:

$$r_i = \sum_{j=1}^{M} w_{ji}(v_j - c_i) \tag{4}$$

$$w_{ji} = \frac{1}{1 + d_{ij}} \tag{5}$$

where $w_{ij}$ is the weight of the residual $v_j - c_i$, which is a decreasing function of the Euclidean distance $d_{ij}$ between $v_j$ and $c_i$. VLAD embedding result is:

$$Z = [r_1 \ldots r_i \ldots r_k] \tag{6}$$

Each small-scale region is inputted into ImageNet-CNN to extract feature vectors with 1000 dimensions of the softmax layer, so each cluster center and VLAD embedding are both 1000-dimensional vectors. In this way, each image obtains a $k \times 1000$ dimensional local representation of the small-scale local regions. We do not use the vectors directly, because the dimensions are too large, so they are not very computationally friendly. Max pooling is conducted on $[r_1, \ldots, r_i, \ldots, r_k]$ to form a 1000-dimensional vector. Finally, we average the results to eliminate the impact of an unbalanced number of features in each image. The final local representation of the small-scale local regions is:

$$V_{\{64,80,96\}} = \frac{1}{M}\text{max} - \text{pooling}\left([r_1, \ldots, r_i, \ldots, r_k]\right) \tag{7}$$

The numbers of feature in small-scale regions extracted for the images are different, which leads to a large difference in the residual of each cluster center. Averaging the results can eliminate this effect.

### 3.3. Multi-Scale Fusion Feature

Multi-scale feature fusion is widely used in scene recognition. Different scales need to be unified in order to fuse. Fusion makes features more robust and easier to learn [49–51]. WS-AM generates many discriminative regions for each image, and multi-scale ($s = 64, 80, 96, 112, 128, 144$) are taken for each discriminative region $p_i$. The form of local regions extracted from each image is:

$$P = [p_1, \ldots, p_i, \ldots, p_N] \tag{8}$$

where $N$ represents the number of local regions. Small-scale regions ($s = 64, 80, 96$) can be considered to contain 'object', so they are inputted to ResNet18 pre-trained on ImageNet (i.e., ImageNet-ResNet18) to extract the 1000-dimensional feature vectors in the softmax layer. The large-scale regions ($s = 112, 128, 144$) which can be considered to contain 'scene', are inputted to ResNet18 pre-trained on Places365 (i.e., Places365-ResNet18) to extract the 365-dimensional feature vectors in the softmax layer. After improved VLAD coding and pooling, we obtain feature vectors $V_{\{64,80,96\}}$ for the small-scale region. Also, we use max pooling to aggregate the features of large-scale regions and obtain the feature vector $V_{\{112,128,144\}}$ for each image. In order to get the global information, we resize each image into the size of $224 \times 224$ and input the entire image into VGGNet pre-trained on Places205 (i.e., Places205-VGGNet) to extract the feature vector $V_{GR}$ of the fc6 layer. We use L2-normalization on $V_{\{64,80,96\}}$ and $V_{GR}$ to obtain $V_{\{64,80,96\}-L2}$ and $V_{GR-L2}$, respectively. L2-normalization is not used on $V_{\{112,128,144\}}$ because the feature vectors are extracted from the softmax layer which can play the role of normalization. Finally, three feature vectors are concatenated to form the final image representation:

$$[V_{GR-L2}\ V_{\{112,128,144\}}\ V_{\{64,80,96\}-L2}] \tag{9}$$

Table 1 shows the tensor dimensionalities in the processing pipeline.

**Table 1.** The tensor dimensionalities in the processing pipeline.

| Tensor | Dimensionality |
|---|---|
| input image | $3 \times 224 \times 224$ |
| feature map | $512 \times 14 \times 14$ |
| gradient | $512 \times 14 \times 14$ |
| weight | $512 \times 1 \times 1$ |
| attention map | $1 \times 14 \times 14$ |
| $V_{GR}$ / $V_{GR-L2}$ | $1 \times 4096$ |
| $V_{\{112,128,144\}}$ | $1 \times 365$ |
| $V_{\{64,80,96\}}$ / $V_{\{64,80,96\}-L2}$ | $1 \times 1000$ |
| $[V_{GR-L2} \; V_{\{112,128,144\}} \; V_{\{64,80,96\}-L2}]$ | $1 \times 5461$ |

*3.4. Classification*

In this paper, linear SVM classifier implementing 'one-vs-the-rest' multi-class strategy is trained on three datasets. Other kernel functions, such as the polynomial kernel, radial basis function (RBF), and sigmoid kernel, are not suitable for our task. Compared with other kernel functions, the linear kernel function has two advantages:

- The linear kernel has fewer hyperparameters and faster training speed.
- The linear function is suitable for high-dimensional features. In this paper, each image has a 5461-dimensional feature vector.

Therefore, we choose linear SVM as the classifier. Penalty parameter $C$, as an important parameter for the SVM model, represents the tolerance of error. Here $C = 1.0$.

## 4. Experiments and Results

The experiments are performed on three datasets: MIT indoor 67, Scene 15 [15] and UIUC Sports [52]. The three datasets contain different types of scene images: MIT indoor 67 mainly contains indoor scene images; Scene 15 contains both indoor and outdoor scene images; UIUC Sports contains event scene images. Then, some parameters of our method are evaluated, including the number of cluster centers, the threshold to extract discriminative regions on AM, different backbone networks of Grad- CAM, and the different scales of the discriminative region. Figure 6 shows some images in the three datasets.

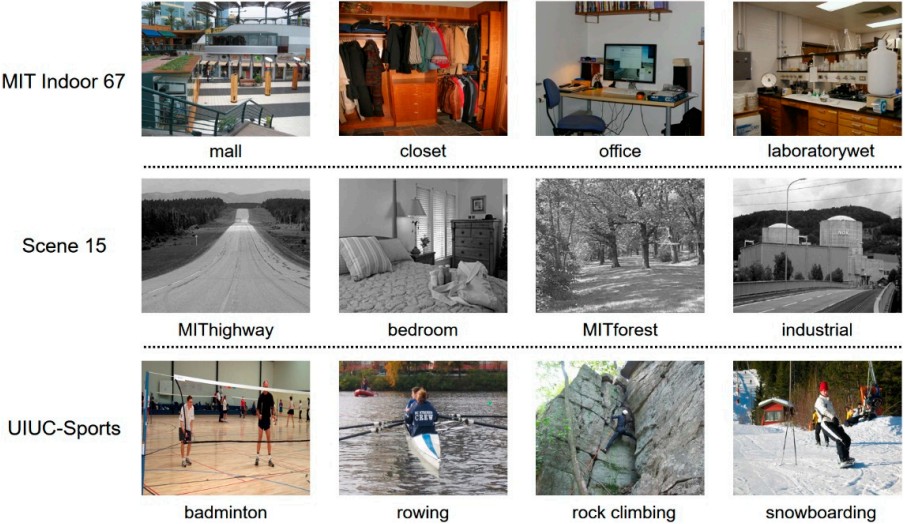

**Figure 6.** Some image examples of the three scene datasets.

### 4.1. Datasets

**MIT indoor 67:** This dataset contains 67 categories of indoor scene images. There are 15620 images in total, with at least 100 images in each category. We follow the division of training set and test set in ref. [10]; 80 images of each category are used for training, and 20 images are used for testing.

**Scene 15:** There are 15 categories in this dataset, a total of 4485 grayscale indoor and outdoor images. The dataset does not provide criteria for dividing the training set and test set. We randomly divide the dataset five times, 100 images of each category are for training, and the rest are used as test images. Finally, we calculated the average accuracy of five times of division.

**UIUC Sports:** This dataset contains eight sports event scene categories, including rowing, badminton, polo, bocce, snowboarding, croquet, sailing, and rock climbing. There are 1579 color images. The dataset does not provide criteria for dividing the training set and the test set. We randomly divide the dataset five times and select 70 training images and 60 test images for each category. Finally, we calculate the average accuracy of five times of division.

### 4.2. Comparisons with State-of-the-Art Methods

MIT indoor 67 dataset mainly verifies the performance in indoor scenes, while Scene 15 verifies the performance both in indoor and outdoor scenes. UIUC Sports verifies the performance in event scenes. The experimental parameters are the same on the three datasets.

Table 2 shows the performance of our method on MIT indoor 67 dataset and its comparison with other methods. The references [1,2,15,53–55] are traditional methods, which mainly use some low-level features and mid-level features, such as SIFT, Object Bank [53], and BOF. Because these features only consider the shape, texture, and color information without any semantic information, they do not have high recognition accuracy. The references [5,6,8,56,57] are based on CNNs, and their overall recognition accuracies are higher than those of traditional methods. The CNN features of scene image have certain semantic information, and these features are learnt from a large number of well-labeled data, while not designed artificially. Our method is remarkably superior to the compared state-of-the-art methods in Table 2, which uses both semantic information and discriminative regions. In addition, the number of local regions used by our method is less than those in other methods, so the overall running time is significantly reduced. Figure 7 shows the confusion matrix of the MIT indoor 67 dataset. We see that the probability of classification is mostly concentrated on the diagonal line, and the overall performance is great. However, some categories have lower recognition accuracy than others, such as 'museum' and 'library' categories. These categories do not work well because the images of these categories are similar to each other and have complex backgrounds.

**Table 2.** Accuracy comparison on MIT indoor 67 dataset.

| Method | Accuracy (%) |
|:---:|:---:|
| SPM [15] | 34.40 |
| CENTRIST [2] | 36.90 |
| Object Bank [53] | 37.60 |
| Discriminative Patches [54] | 38.10 |
| OTC [1] | 47.33 |
| FV + Bag of parts [55] | 63.18 |
| Places-CNN [56] | 68.24 |
| Hybrid-CNN [56] | 70.80 |
| Semantic FV [6] | 72.86 |
| MetaObject-CNN [8] | 78.90 |
| VS-CNN [5] | 80.37 |
| LS-DHM [57] | 83.75 |
| Our WS-AM {64,80,96,112,128,144} | 81.79 |
| Our WS-AM {64,80,96,112,128,144} + fc6 features | 85.67 |

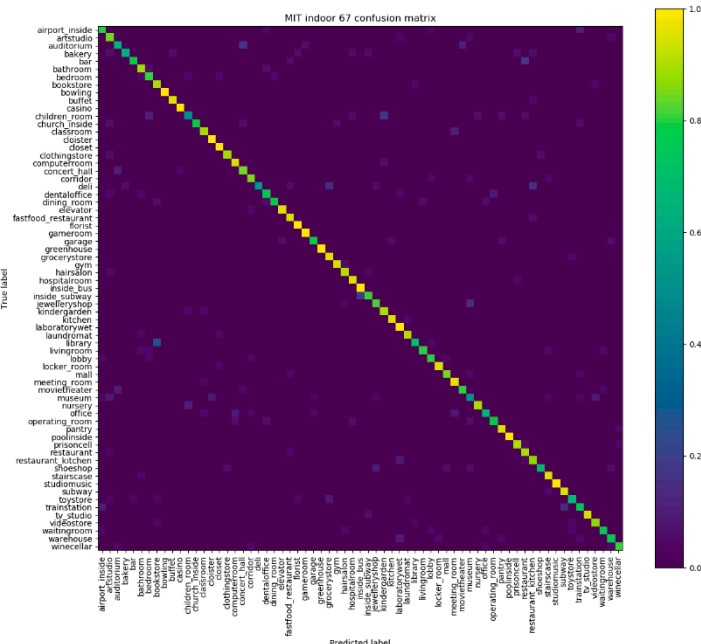

**Figure 7.** Confusion matrix of MIT indoor 67 dataset.

The experiments are carried out on Scene 15 dataset, which contains both outdoor and indoor scenes. Table 3 tabulates the comparison results in Scene 15 dataset. Our method achieves the recognition accuracy of 94.80% and is markedly superior to the compared state-of-the-art methods. Figure 8 shows the confusion matrix of Scene 15 dataset. The accuracy of the 'CALsuburb' class reaches 100%. The accuracies of 'MITcoast', 'MITforest', 'MIThighway', and 'MITmountain' categories is very high, and it can be concluded that our method also performs well in outdoor scenes. However, it can be clearly seen from the confusion matrix that the accuracies in outdoor scenes are relatively lower than those in indoor scenes.

**Table 3.** Accuracy comparison on Scene 15 dataset.

| Method | Accuracy (%) |
|---|---|
| LDA [15] | 59.00 |
| BoW [15] | 74.80 |
| Object Bank [58] | 80.90 |
| SPMSM [59] | 82.30 |
| OTC [1] | 84.37 |
| Places-CNN [56] | 90.19 |
| Hybrid-CNN [56] | 91.59 |
| DGSK [60] | 92.30 |
| Our WS-AM {64,80,96,112,128,144} | 92.58 |
| Our WS-AM {64,80,96,112,128,144} + fc6 features | 94.80 |

Table 4 tabulates the comparison results on UIUC Sports dataset. Our method achieves an accuracy of 95.12% and is superior to the compared state-of-the-art methods. UIUC Sports is a dataset of sport event scenes, which is different from the general scenes. The confusion matrix of the UIUC Sports dataset is indicated in Figure 9. We see that the recognition accuracy of the 'sailing' category reaches 100%, and the accuracies of the classes except 'bocce' and 'croquet' are good. It is because the contents of these two scene categories are similar, e.g., 'people' and 'ball'.

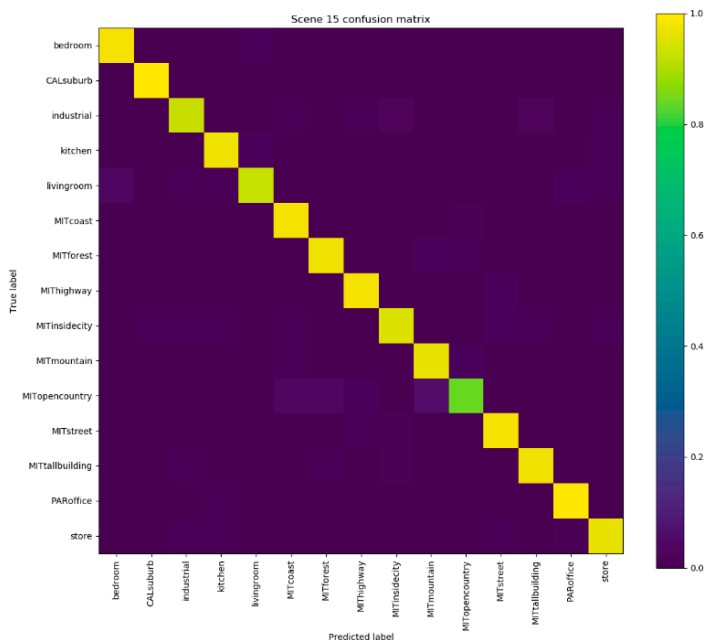

**Figure 8.** Confusion matrix of Scene 15 dataset.

**Table 4.** Accuracy comparison on UIUC Sports dataset.

| Method | Accuracy (%) |
| --- | --- |
| GIST-color [61] | 70.70 |
| MM-Scene [62] | 71.70 |
| Object Bank [53] | 76.30 |
| CENTRIST [2] | 78.25 |
| SPMSM [59] | 83.00 |
| DF-LDA [7] | 87.34 |
| VC + VQ [63] | 88.40 |
| ISPR + IFV [64] | 92.08 |
| Our WS-AM {64,80,96,112,128,144} | 93.07 |
| Our WS-AM {64,80,96,112,128,144} + fc6 features | 95.12 |

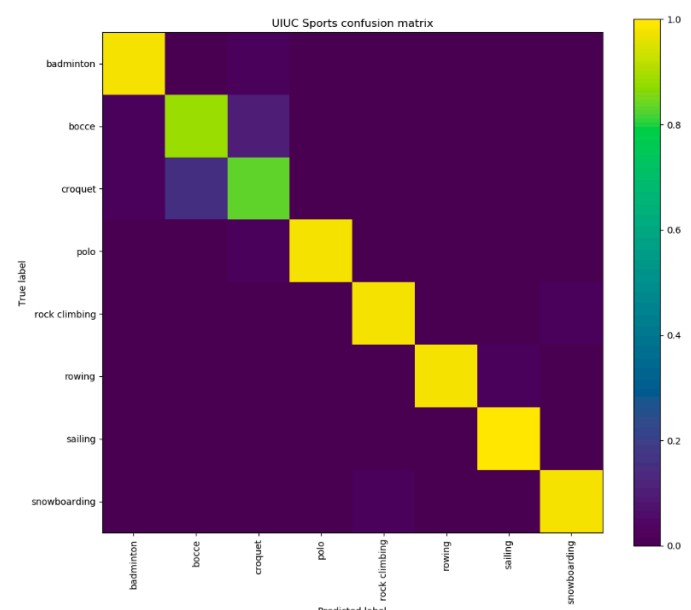

**Figure 9.** Confusion matrix of UIUC Sports dataset.

## 5. Experiments Analysis

In this section, we evaluate several important parameters of our method. First, we compare the performance of different backbone network structures for the Grad-CAM method. Second, we evaluate the impact of different scale combinations of discriminative regions on the results. Third, the effect of different thresholds is evaluated. Fourth, the number of cluster centers is very important for the aggregation of local features, so the influence of a different number of cluster centers is compared. Fifth, we prove the importance of L2-normalization. Sixth, we compare the performance of the different parameter $C$. Finally, in order to demonstrate the effectiveness of the WS-AM method for obtaining discriminative regions of scene images, we visualized discriminative regions of some categories. All of these evaluations are performed on MIT indoor 67 dataset.

### 5.1. Evaluation

**Backbone network.** In our WS-AM method, VGGNet pre-trained on Places205 dataset (i.e., Places205-VGGNet) from ref. [65] is used as the backbone network to obtain AM. Three pre-trained networks are evaluated including VGG11, VGG16, and VGG19. Table 5 lists the recognition results of three backbone networks on the MIT indoor 67 dataset. It shows that VGG11 performs better than the other networks, and its accuracy is 2.17% (1.72%) higher than that of VGG19 because the discriminative regions extracted from VGG11 are more representative. On the other side, the VGGNet is also used to extract the global feature in fc6 layer for each image, so the final recognition accuracy is affected by two factors: discriminative regions and global features.

**Table 5.** Performance of different backbone networks on MIT indoor 67 dataset.

| Network | Accuracy (%) (+fc6 Features) |
|---------|------------------------------|
| VGG11   | 81.79 (85.67)                |
| VGG16   | 80.07 (84.25)                |
| VGG19   | 79.62 (83.95)                |

**Scale.** Six rectangular regions of different scales ($s = 64, 80, 96, 112, 128, 144$) are cropped for each discrimination region and the performances of different scale combinations are compared on MIT indoor 67 dataset. The regions at the scales of ($s = 64, 80, 96$) contain 'object', while the regions at the scales of ($s = 112, 128, 144$) contain 'scene', so these regions with two different scales are inputted into different CNNs to extract features in the softmax layer. Table 6 indicates the influence of different scale combinations on recognition accuracy. We see that the scales of ($s = 64, 80, 96, 112, 128, 144$) perform better than other combinations of scales because the objects in the scene are basically multi-scale, and we can obtain features containing more scale information by using more scales. On the one hand, from rows 1–3, 4–6, and 7–9 in Table 6, it can be seen that the coarse local scales ($s = 112, 128, 144$) are important to extract global information. On the other hand, from rows 2, 5, 8, and 3, 6, 9 in Table 6, we can see that the fine local scales ($s = 64, 80, 96$) are significant to extract local information.

**Table 6.** Performance of different scales on MIT indoor 67 dataset.

| Scale | Accuracy (%) (+fc6 Features) |
|-------|------------------------------|
| 64, 112 | 78.65 (83.50) |
| 64, 112, 128 | 80.37 (84.32) |
| 64, 112, 128, 144 | 80.74 (84.77) |
| 64, 80, 112 | 79.02 (83.95) |
| 64, 80, 112, 128 | 80.82 (84.25) |
| 64, 80, 112, 128, 144 | 81.19 (84.92) |
| 64, 80, 96, 112 | 80.00 (84.32) |
| 64, 80, 96, 112, 128 | 81.26 (84.62) |
| 64, 80, 96, 112, 128, 144 | 81.79 (85.67) |

**Threshold.** Two strategies are used in Section 3 to screen the discriminative regions in AM. For the first strategy, different thresholds (0, 50, 100, 150) are experimented on MIT indoor 67 dataset and its impact evaluated on the recognition results. From the results in Figure 10, we see that the recognition accuracy is the highest when the threshold is 100 and the lowest when the threshold is 150 (without fc6 features). This indicates that more discriminative regions will improve the performance of the recognition, and fewer regions will result in a lack of local information. However, when the global features (fc6 features) are combined, the recognition accuracy is the highest when the threshold is 100, which is 85.67%. It is because the threshold only affects local representations, and when combined with global features, the overall trend will change. In this paper, 50 spacing is used to evaluate the threshold without considering smaller spacing. In future work, further optimization may lead to performance improvement.

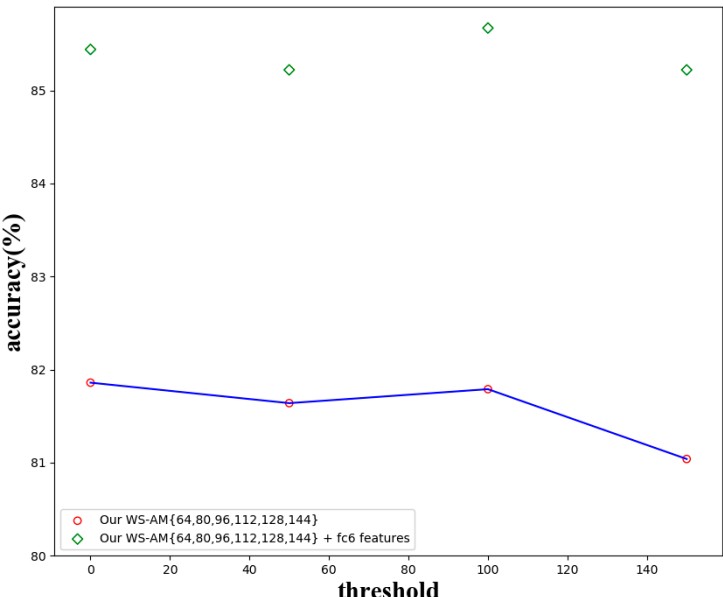

**Figure 10.** The recognition accuracies of different thresholds on MIT indoor 67 dataset.

**Cluster center.** To evaluate the impact of a different number of cluster centers, an experiment on MIT indoor 67 dataset is carried out with a various number of cluster centers. Figure 11 shows the effects of a various number of cluster centers. It can be seen that when the number of centers is 40 and 70, the recognition accuracy is the highest (82.23% without fc6 features). The unreasonable number of cluster center leads to poor generality, and further, degrades accuracy. However, combined with the global features (fc6 features) the recognition accuracy reaches 85.67% when the number of centers is 10 because the VLAD centers only affect local representations.

**L2-normalization.** Normalization is the process of scaling individual samples to have unit norm. After normalization, features are easier to be trained by SVM, which means it is easier to find the classification hyperplane of features. If the features are not normalized, SVM may not converge because the numerical range of each dimension is different. In this paper, features are normalized with L2-normalization. Table 7 shows the accuracy with L2-normalization or without L2-normalization on the MIT indoor 67 dataset. We can see that the feature with L2-normalization achieves better results. However, when $V_{\{112,128,144\}}$ is normalized, the accuracy is reduced by 0.97%. It is because the feature vectors are extracted from the softmax layer which can play the role of normalization.

**Parameter** $C$. Penalty parameter $C$ is an important parameter for the SVM model. $C$ represents the tolerance of error. When the parameter $C$ is large, the SVM model will be over-fitting. Therefore, a suitable $C$ will bring better results to the recognition. Table 8 shows the accuracy of the different $C$ on MIT indoor 67 dataset. It can be seen that with the increase of $C$, the accuracy will decline.

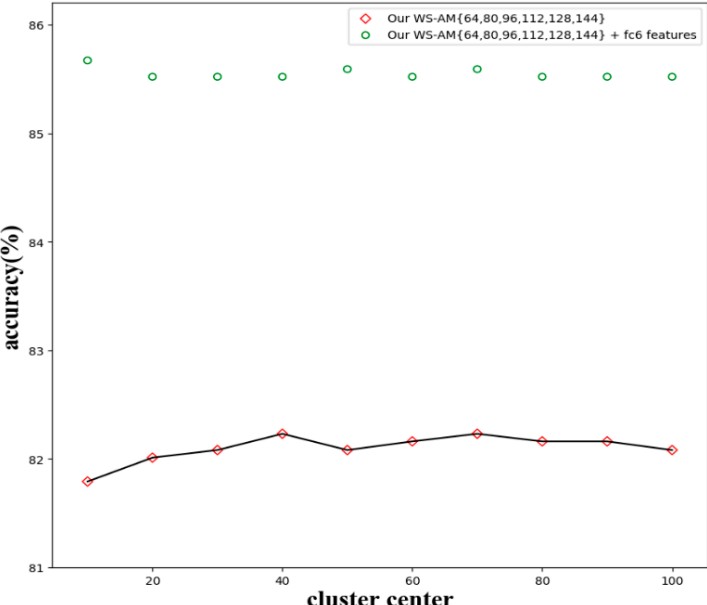

**Figure 11.** The recognition accuracies of a different number of cluster centers on MIT indoor 67 dataset.

**Table 7.** Accuracy with L2-normalization or without L2-normalization on MIT Indoor 67 dataset.

| $V_{GR}$ | $V_{\{112,128,144\}}$ | $V_{\{64,80,96\}}$ | Accuracy (%) |
|:---:|:---:|:---:|:---:|
| ✗ | ✗ | ✗ | 81.86 |
| ✔ | ✔ | ✔ | 84.70 |
| ✔ | ✗ | ✔ | 85.67 |

[7] ✗: without L2-normalization; ✔: with L2-normalization.

**Table 8.** Accuracy of the different parameter *C* on MIT indoor 67 dataset.

| Parameter *C* | Accuracy (%) |
|:---:|:---:|
| 1 | 85.67 |
| 2 | 84.55 |
| 3 | 84.40 |
| 4 | 84.17 |
| 5 | 84.02 |

### 5.2. Visualization of Discriminative Regions

In order to demonstrate that the extracted regions are discriminative, we visualize some discriminative regions of four scene categories ('nursery', 'museum', 'croquet', 'industrial') from different datasets. In Figure 12, we show some examples of discriminative regions from four categories. The discriminative regions correspond to the visual mechanism of human observation scenes, for examples, a baby's cot in a nursery, a ball club on a court, and a painting of a museum. This indicates that the discovered regions contain the objects specific to the context of the scene image, and they are helpful to scene recognition.

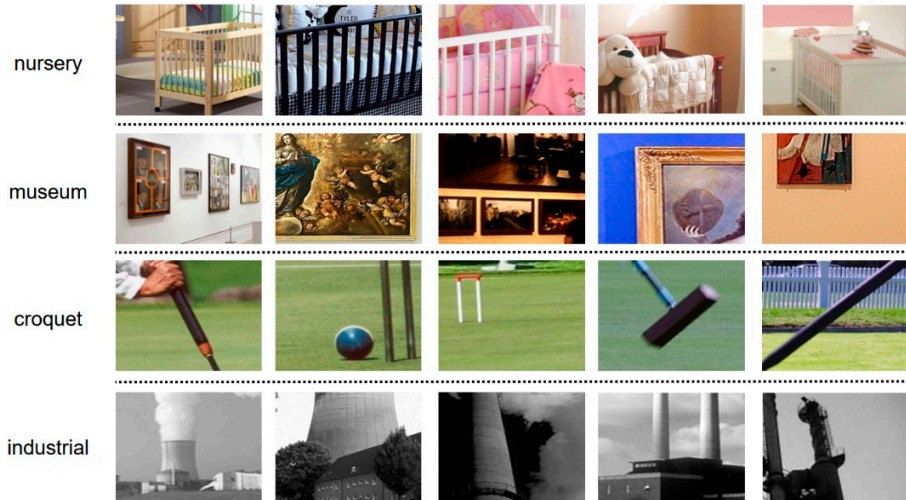

**Figure 12.** Examples of discriminative regions discovered by our WS-AM method.

## 6. Conclusions

In this paper, we proposed a WS-AM method to discover discriminative regions in scene images. Combined with the improved VLAD coding, we could extract more robust features for scene images. Compared with existing methods, our method selects fewer local regions containing semantic information to avoid the influence of redundant regions. The improved VLAD coding is more suitable for our method than the general VLAD coding. The experiments were carried out on three benchmark datasets: MIT indoor 67, Scene 15, and UIUC Sports, and obtained better performance. Our work was inspired by fine-grained image recognition, whose main task was to find the discriminative regions within the class. In the future, we will improve our methods and apply them to other recognition tasks.

**Author Contributions:** Conceptualization, data curation and formal analysis, S.X.; Funding acquisition, J.Z.; Investigation, S.X and L.L.; Methodology, S.X.; Project administration, S.X and X.F.; Resources, J.Z. and L.L.; Software, S.X.; Supervision, J.Z. and X.F.; Validation, visualization and writing—original draft, S.X; Writing—review & editing, J.Z, L.L and X.F.

**Funding:** This work was supported in part by the National Natural Science Foundation of China under Grant 61763033, Grant 61662049, Grant 61741312, Grant 61866028, Grant 61881340421, Grant 61663031, and Grant 61866025, in part by the Key Program Project of Research and Development (Jiangxi Provincial Department of Science and Technology) under Grant 20171ACE50024 and Grant 20161BBE50085, in part by the Construction Project of Advantageous Science and Technology Innovation Team in Jiangxi Province under Grant 20165BCB19007, in part by the Application Innovation Plan (Ministry of Public Security of P. R. China) under Grant 2017YYCXJXST048, in part by the Open Foundation of Key Laboratory of Jiangxi Province for Image Processing and Pattern Recognition under Grant ET201680245 and Grant TX201604002, and in part by the Innovation Foundation for Postgraduate Student of Nanchang Hangkong University under Grant YC2018095.

**Conflicts of Interest:** The authors declare no conflict of interest.

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
