# Peer review of "WS-AM: Weakly Supervised Attention Map for Scene Recognition"

_electronics, doi:10.3390/electronics8101072_

Round 1

Reviewer 1 Report

In this paper, the authors tackled the scene recognition problem using deep learning-powered techniques coupled with attention mechanisms. The manuscript is well-written and the results are promising, however it suffers from several issues which need to be resolved:

The authors should properly highlight their contribution (determining the important parts of an input scene using attention modules, and extracting discriminative features from them). There exist a number of works which use attention modules for various tasks, just to mention a couple of them: https://arxiv.org/abs/1411.6447 https://arxiv.org/abs/1811.02667 https://arxiv.org/abs/1706.03762 https://arxiv.org/abs/1603.06765 https://ieeexplore.ieee.org/document/8099959 https://arxiv.org/pdf/1808.00677.pdf

These papers should be discussed in the related work, the authors should clearly discuss the differences and similarities between the attention modules used. Therefore, the authors failed to contextualize their work within the state of the art concerning deep networks enhanced with attention maps.

2. Although I do have (fairly extensive) experience in deep learning, the description of the proposed method is a bit confusing, especially in the context of attention maps. I suggest to provide a simple end-to-end example of extracting features from input scenes using attention to allow a reader clearly understand it (Figure 3 is a good start, however it is not very informative). Also, I suggest adding a figure showing the tensor dimensionalities in the processing pipeline for an example input scene.

3. What do the authors mean by saying that "discriminative model does not well reflect the characteristics of the data"? This should be discussed in more detail.

4. The motivation behind some decisions remains unclear. The first paragraph of Sect. 3 should shed more light on that. Also, why did the authors select L2 normalization? Why a linear SVM has been selected as a classifier? How the C parameter was selected? Was it tuned in any way?

5. The experiments seem to be not reproducible, especially when we consider the data splits, e.g., for Scene 15. The authors mentioned that they divided the set five times -- was it non-overlapping cross-validation? I suggest making their divisions publicly available to allow other research groups reproduce their work. Also, I encourage the authors to make their code open-sourced.

Reviewer 2 Report

This paper proposed a weakly supervised activation mapping (WS-AM) approach to discover discriminative regions in images. And combined with VLAD coding, they extracted better features for scene recognition.

The proposed model is reasonable and lean to accept the paper, but I have the following questions:

The proposed model contains several CNNs, as indicated in Figure 3, Places-CNN, ImageNet-CNN. Then is it still a fair comparison with baseline models?  The models could give wrong attentions, so it will be informative to give failure examples. What will happen if the attention is wrong?

Round 2

Reviewer 1 Report

I appreciate seeing that the authors addressed most of my concerns. However, two remain unresolved:

It is still unclear why support vector machines with linear kernels have been selected for comparison. Why not e.g., radial-basis functions? It should be discussed in the paper. It seems that the authors did not discuss the attention modules being applied to other modalities, and did not compare their modules with them: https://arxiv.org/abs/1706.03762 https://arxiv.org/abs/1811.02667 I believe it may help give a reader a wider overlook of attention-based applications in the literature.
